

# Role of tea plantations in the maintenance of bird diversity in Anji County, China

Jueying Wu, Jinli Hu, Xinyu Zhao, Yangyang Sun and Guang Hu

School of Civil Engineering and Architecture, Zhejiang Sci-Tech University, Hangzhou, China

## ABSTRACT

**Background.** Tea plantations support regional sustainable development and have the potential to support more biodiversity than urban open spaces. Numerous studies have shown the value of low-intensity agroecosystems for preserving biodiversity, however tea plantations have received less attention. The relationship between tea plantations and the diversity of macro-organisms, such as birds, is still not fully understood.

**Methods.** We investigated the bird diversity and vegetation conditions and calculated landscape metrics in 30 tea plantations in Anji County, Zhejiang Province, China. At these 30 sampling sites, we recorded 262 individuals belonging to 37 species, which were classified into two guilds: nature- and urban-dependent birds. We used cluster analysis to group the sampling sites based on the abundance of the birds. Then we evaluated the effects of associated plant diversity in tea plantations and the surrounding landscape composition on these bird guilds using species association computation and a generalized linear model.

**Results.** The results show that the maintenance of bird diversity by tea plantations benefits both nature- and urban-dependent birds. We found that landscape-scale factors surrounding the tea plantations mainly affected the bird richness due to their habitat selection. Landscape agglomeration and habitat quality were the dominant landscape-scale metrics. Patch-scale factors of tea plantations, especially the vegetation structure, had a strong influence on the abundance of the birds. Nature-dependent birds preferred to occur in tea plantations with perennial herbs, while urban-dependent birds were attracted by the general distributed plants, as annual herbs. Therefore, we concluded that tea plantations play an important role as a transitional zone between natural habitats and urban areas, thus reducing the impact of urbanization and maintaining bird diversity in low-quality habitats.

## INTRODUCTION

Tea (*Camellia sinensis*) is one of the main beverage plants appreciated for its aroma and beneficial effects on human health. It has been widely cultivated in China for millennia, particularly in tropical and subtropical areas. In 2021, the total area of tea plantations in China was approximately 3.26 million ha, with an output value of 45.4 billion dollars (*Mei & Liang, 2022*); thus, tea planting can be used to effectively to promote the local economy, boost farmers' incomes, and support sustainable regional construction. Like most cash crops, tea plantations are mainly located in the suburbs. However, with the

Corresponding author
Guang Hu, hug163@163.com

recent expansion of construction land and rise of urban agriculture, the distance between cities and tea plantations has been decreasing (*Fu & Liu, 2015*). Some tea plantations near the city or on the rural–urban fringe have been transformed into unique landscapes of greenery that are popular recreational spaces for city dwellers (*Sun & Yang, 2016*), examples include Longjing Tea Village in Hangzhou (*Zhu, Gao & Zhang, 2017*), Huanglong Xian Tea Culture Village in Nanjing (*Yan & Lu, 2021*), and Laoshan Tea Plantation in Qingdao (*Ding, 2015*).

As a land-use type with production and landscape values, tea plantations can support greater biodiversity than urban open spaces (*Lin & Fuller, 2013*). Although tea plantations are referred to as "green deserts" by some natural conversationalists, tea plantations and their surrounding areas in urban areas still have a relatively high level of biodiversity. More than 1,100 natural enemies of agricultural pests, including approximately 40 species of fungi, 240 species of parasitoids, and 600 species of predators, as well as several species of bacteria, have been recorded in tea ecosystems in China (*Ye et al., 2014*). Spiders are the dominant predators of pests in tea plantations, with 535 species in 40 families reported from Chinese tea plantations (*Song et al., 2020*). It has been proven that plantation, when established on degraded lands rather than replacing natural ecosystems, such as forests, grasslands, and shrublands, are most likely to contribute to biodiversity (*Bremer & Farley, 2010*). Tea plantations also provide great value and potential for biodiversity conservation; however, their specific role in biodiversity conservation and maintenance has not been adequately reported. Studies on other beverage plants, such as coffee and cacao plantations, have found that these agroecosystems can be regarded as complementary systems for biodiversity conservation. A study in some remaining forest fragments in Ethiopia revealed that coffee farms could support a considerable portion, of the woody biodiversity of disappearing forests (*Tadesse, Zavaleta & Shennan, 2014*). The UN Environment Programme World Conservation collated original biodiversity field data from 36 studies (1,295 sites) from the cacao-producing regions of the world, and found that incentivizing planted shade agroforestry can enhance the biodiversity intactness in degraded areas while delivering co-benefits (*Maney, Sassen & Hill, 2022*).

Birds are one of the indicators of biodiversity, and their relationship with artificial plantations has been studied (*Castaño Villa et al., 2019*). In most cases, tree plantations play a positive role in support of bird diversity in artificial landscapes. In southern US, it was reported that one- or two-year eucalyptus plantations supported bird diversity similar to those in six- to seven-year pine stands (*Messick et al., 2021*). The combination of different types of vegetation, such as that of the landscape mixed with non-native plantations and native forests, was able to support a higher bird diversity (*Rodríguez-Pérez, Herrera & Arizaga, 2018*). Systematic grazing can improve bird diversity in oil palm plantations by increasing habitat complexity (*Tohiran et al., 2019*).

Although it is a tree species originally, tea is mostly cultivated as a perennial shrub. It is pruned only during harvest to pluck small parts of the fresh growth without damaging the entire plant. Compared to annual crops, it can be considered a relatively stable habitat with the potential to support high biodiversity (*Wordley et al., 2015*). Previous studies on tea plantations have mostly focused on the relationships between tea plantation management,

associated small organisms, and tea yield. Studies have been carried out on higher native plant diversity in traditional organic tea plantations (*Chowdhury, Samrat & Devy, 2021*), the benefits of soil microbial communities in tea plantations (*Huidrom & Sharma, 2014*), theoretical and applied aspects of organic fertilizers for improving the diversity of soil microbial communities in tea plantations (*Gu et al., 2019*), and pest control by pruning intensity in tea plantation management (*Wang, Ma & Li, 2015*). Although these studies have promoted the development of eco-tea plantations, the correlations between tea plantations and macro-organisms, such as birds, mammals, and vascular plants, are still not fully understood. Tea-dominated landscapes can support many forest-associated bat and bird species because they contain forest fragments and shaded areas (*Imboma et al., 2020*). As part of a fragmented landscape, yerba mate and tea are perennial agroecosystems that provide habitat continuity for diverse arthropods (*Rubio et al., 2019*). In a previous study, suitable habitats for tea plant cultivation and areas where these habitats overlap with the Asian elephant distribution to balance the conflict between the expansion of tea plantations and the spatial needs of wildlife in southwestern China were identified (*Dai, 2022*). In another study, windbreaks were placed in the tea plantations in the Western Ghats to enhance the richness and abundance of ecological service providers such as frugivores, insectivores, and nectarivores against extensive species loss in tea plantations to sustain bird diversity in a tea-dominated landscape (*Sreekar et al., 2013*).

Anji County, located in Zhejiang Province in East China, is known as the 'White Tea Capital'. Tea production is an essential source of economic revenue in Anji. In 2021, tea plantations occupied an area of over 133 km$^2$ and yielded more than 2000 tons of tea; the primary production value of this tea was 482.2 million dollars, accounting for more than 60% of the county's total agricultural output. Here, we investigated the use of tea plantations by birds. We compared the species richness and abundance of birds in Anji County, and we evaluated the potential contribution of tea agroecosystems to the maintenance of affiliated bird communities. Specifically, we investigated the landscape composition and locally associated plant diversity in 30 tea plantation systems located in Anji to assess the effects of landscape- and local-scale factors on the bird community and bird guilds, namely urban- and nature-dependent birds. Our goal was to estimate the role of tea plantations in conserving urban bird biodiversity and provide a theoretical basis and database for future ecological, sustainable development of tea plantations. In this way, we can also identify which cultivation and land-use practices promote bird diversity conservation in tea plantations and surrounding areas.

## MATERIALS & METHODS

### Study sites

This study was conducted in Anji County, Zhejiang Province, China (119°14′−119°53′E, 30°23′−30°53′N). The terrain is characterized by large and chained mountains, along with small and scattered plains, with a 75% vegetation cover and diverse terrestrial ecosystems. Anji has a subtropical monsoon climate, with an average annual precipitation of 1,100–1,900 mm and an average annual temperature of 16−19 °C. The main soil type is acidic
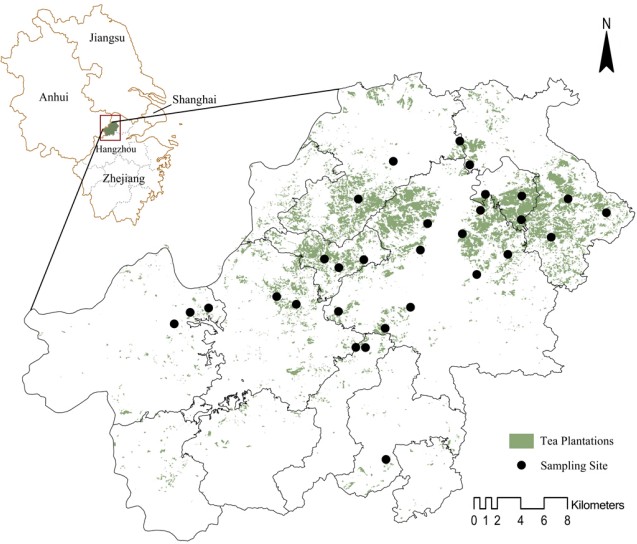

**Figure 1** **Locations of sampling sites in Anji County.** Data source credit: Standard Map Service System, Ministry of Natural Resources (GS (2020)4619, http://bzdt.ch.mnr.gov.cn/).

red, which is suitable for growing tea trees. By the end of 2020, the number of existing white tea plantations in Anji was 11,813, forming a spectacular tea plantation landscape. Tea plantations are distributed throughout Anji County and the main production areas. A total of 30 sampling sites in the most concentrated areas of tea plantations in Anji County were selected for the survey. Each sampling site was at least 500 m away from the others (Fig. 1).

## Bird Survey

We surveyed the bird communities at the 30 selected sampling tea plantations using point counts with unlimited distance in August 2021 (*Buckland et al., 2004*). We recorded the number and species identity of birds seen or heard in these plantations, except for domestic birds such as pigeons (Table S2). Each visit lasted 10 min, with an extra 5 min after the observer arrived to give the birds time to settle in. The survey was done during morning hours while we avoided adverse weather conditions such as strong winds and heavy rain (*Bibby, Burgess & Hill, 2000*). Bird surveys were repeated three times at different times. We performed species identification according to the CNG Field Guide to the Birds of China (*Chen & Liu, 2021*), and we quantified species diversity using species richness and abundance. The richness was determined as the total number of observed species at each sampling site and the abundance was determined as the total number of observed individuals.

## Vegetation survey

We employed a random systematic sampling design and constructed 10 m $\times$ 10 m quadrats within each sampling site to census the vegetation (*Tessarolo et al., 2014*). We measured the plant composition and abundance in the plots and classified each species into one of

**Table 1  Landscape metrics of the sampling sites at patch and landscape scales.**

| Scale | Variable | Unit | Description |
|---|---|---|---|
| Patch scale | Slope | ° | Average slope. |
| | E-W Aspect | / | Sine function of slope orientation. |
| | N-S Aspect | / | Cosine function of slope orientation. |
| | Area | $m^2$ | Area of the tea plantation patch where the sample site was located. |
| | Altitude | m | Average altitude. |
| | Plant | species | Number of species in the plot. |
| Landscape scale | PD | patch/ha | Number of patches per unit area in the buffer zone. |
| | LPI | % | Area share of the largest patches in the buffer zone. |
| | CONTAG | % | Degree of agglomeration or extension of patch types in the buffer zone. |
| | Tea | % | Proportion of tea plantation patches in total patch area in the buffer zone. |
| | Quality | $m^2$ | Weighted average of the areas of different types of land around the tea plantation patches in the buffer zone. |

six functional types (trees, shrubs, lianas, vines, annual and biennial herbs, and perennial herbs). We consulted iPlant.cn (https://www.iplant.cn/), a web page based on the Flora of China and Chinese plant image database for the identification of the plant species (Table S3).

## Landscape metrics

Referring to previous studies that investigated the influence of environmental factors on bird diversity in human-modified landscapes, we selected 11 landscape metrics at the patch and landscape scales (Table 1). We extracted patch-scale metrics from DEM data with 5 m precision of Zhejiang Province to fit the raster data of tea plantation distribution in Anji County in 2020, including slope, area, altitude, slope direction, and the number of plant species from the vegetation survey. We downloaded the land use/land cover data for 2021 based on the 10 m resolution Sentinel-2 satellite imagery (*Karra et al., 2021*). Then we calculated the landscape-scale metrics using ArcGIS 10.5 and Fragtats 4.2 within a specific 400 m radius buffer of each site to maximize the coverage of the surrounding landscape while avoiding the overlap. We obtained the habitat quality index by weighting the area of different land use types within the buffer zone, where the weights of water bodies, forest land, cropland, grassland, and construction land were 5, 4, 3, 2, and 1, respectively. These weight scores were assigned with reference to the accessibility of food and water sources for birds (*Sohil & Sharma, 2020*).

## Data analysis

We collated information on the distribution and dietary traits of each bird species recorded using a Chinese avifauna (*Zheng, 2017*) and a dataset of bird traits (*Wang et al., 2021*), as well as our field observations. To determine the different responses of bird guilds to different characteristics of tea plantations, we classified the recorded bird species into two guilds: nature- and urban-dependent (Table S1), according to their ordinary distribution

(*Chen, 2000*) and their general occurrence on different land use types (*Wang, Chen & Ding, 2004*). Those birds classified as nature-dependent (ND) preferred habitats such as ponds, mountains, forests and other natural land types, as mentioned in previous studies, while urban-dependent birds (UD) were recorded in built-up areas, parks, and other urban artificial environments (*Wang, Chen & Ding, 2004*).

The locations of sampling sites (Fig. 1) are shown on the administrative map from the Standard Map Service System, Ministry of Natural Resources (No.GS(2020)4619, http://bzdt.ch.mnr.gov.cn/). All statistical analyses were applying in R 4.0.3 (*R Core Team, 2020*). We constructed a dendrogram based on the composition of bird guilds at the 30 sampling sites using Bray–Curtis distance and Ward cluster by the "Vegan" package. Afterward, these analyses were used to categorize the tea plantations and determine the relationship between plants and different bird guilds in the tea plantations. Analysis of variance (ANOVA) was applied with the "stats" package to test the significant differences between the associate plant compositions across these categories of tea plantations, and referring to $p \leq 0.05$ probability level. We used the "Spaa" package to calculate species association (*Zhang, 2016*). The associated strength was assessed by these general guidelines (*Cohen, 1988*). When the absolute value of association $r$ was more than 0.3, it meant a strong association. To evaluate how environmental factors influenced bird richness and abundance, we constructed a Poisson-linked generalized linear model (GLM) by using the "Lme4" package (*Douglas et al., 2015*) and included 11 landscape metrics (Table 1) as the explanatory variables. This model was also applied to each bird guild (nature-dependent and urban-dependent). For testing the multicollinearity, we checked the variance inflation factor (VIF) of each metric using the "car" package, which all did not exceed 10. Then a stepwise analysis was applied to the basic models to find the best-fit models for all birds and for each guild. Variable selection and model evaluation were both based on AICc (*Buonaccorsi, 2011*). The variable's significance was adjusted using Bonferroni correction. In addition, we used hierarchical partitioning to assess the relative importance of these variables with a randomization test using the "Hier.part" package (*Walsh & Nally, 2004*, Table S5).

## RESULTS

### Bird composition in tea plantations

We recorded 262 birds at the 30 sampling sites, classified into 23 families and 37 species. The families Ardeidae (three species), Accipitridae (three species), Corvidae (three species), and Muscicapidae (three species) had the highest species richness, and the families Hirundinidae ($n = 46$), Columbidae ($n = 40$), and Passeridae ($n = 33$) had the highest abundances. The most abundant species recorded was *Passer montanus* ($n = 33$), followed by *Pycnonotus sinensis* ($n = 30$) and *Hirundo rustica* ($n = 29$). A total of 14 urban-dependent bird species were observed, with 64% of the total abundance. These birds were better adapted to the artificial environment and more commonly encountered in cities. A total of 23 nature-dependent bird species were recorded. This bird guild preferred more natural habitats, including most migrant and rare birds.

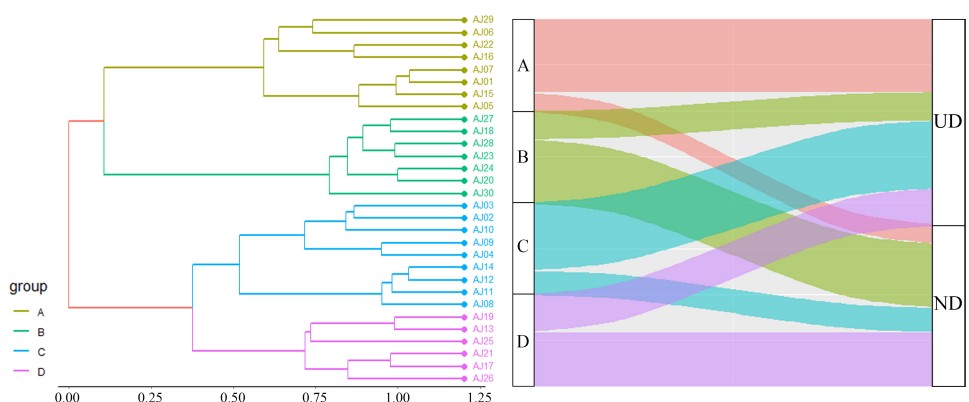

**Figure 2** **Cluster dendrogram of habitat types with bird guild composition.** UD, urban-denpendent birds; ND, nature-dependent birds; (A-D) bird clusters based on species composition.

## Association between bird and plant composition

A cluster dendrogram of sampling sites based on the bird species composition showed four clusters (Fig. 2): A (eight sites), B (seven sites), C (nine sites), and D (six sites). The tea plantations in clusters A and C were dominated by UD birds, while those in clusters B and D were mainly ND birds. Clusters A and C were close to residential areas occupied mostly by generalist species such as *Pycnonotus sinensis* and *Spilopelia chinensis*, while clusters B and D comprising mixed habitats supported a wide variety of species, including *Corvus torquatus*, *Bambusicola thoracicus*, and *Bubulcus coromandus*.

The ANOVA results for the plant composition indicated significant differences in the types of companion plants among the four tea plantation clusters ($p = 0.02$) (Fig. 3). The types with the highest proportion of plants in clusters B and D were perennial herbs, such as *Arthraxon prionodes*, *Hedyotis chrysotricha*, *Cyperus rotundus*, and *Lysimachia fortunei*. Clusters C and D were shared mostly by annual and biennial herbs, including *Lindernia crustacea*, *Digitaria sanguinalis*, *Crassocephalum crepidioides*, and *Euphorbia maculata*. Species association analysis between plant composition and bird guilds (Fig. 4) showed that nature-dependent birds mainly preferred the surviving trees in the plantation, some shrubs and perennial herbs, and were weakly negative with lianas and annual and biennial herbs. UD birds were significantly positive associated with shrubs and annual and biennial herbs and had a weakly positive association with trees and vines. Both ND and UD birds were recorded in the tea plantation with woody plants (trees and shrubs), while annual and biennial herbs had opposite associations on these guilds.

## Influence of landscape attributes on the bird diversity

The results of best-fit GLM show that landscape-scale metrics had a stronger impact than patch-scale metrics on the bird diversity in the tea plantations (Fig. 5, Table 2). At the landscape scale, bird richness and abundance decreased with the increase in habitat quality and CONTAG metrics. More factors significantly affected the abundance of each guild than richness, especially the patch-scale metrics, which did not affect the richness. ND birds were predominantly influenced by landscape-scale factors, while UD birds

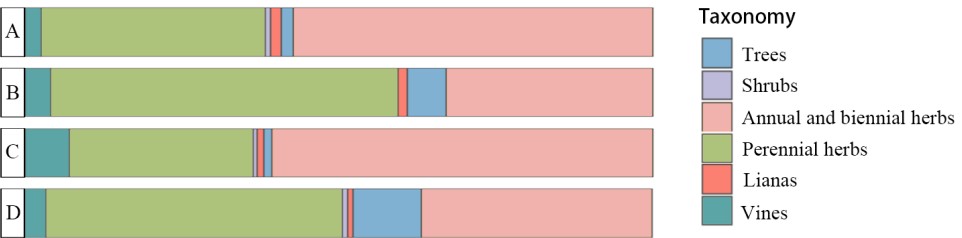

**Figure 3** **Associate plant composition of the four tea plantation types.**

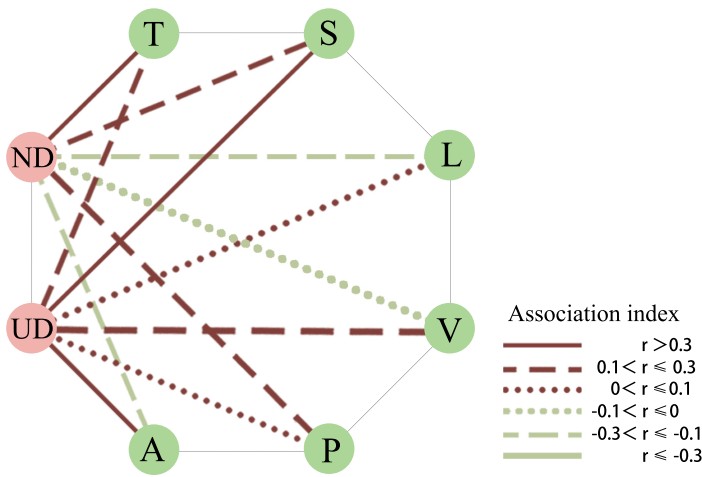

**Figure 4** **Species association network.** ND, nature-dependent birds; UD, urban-dependent birds; T, trees; S, shrubs; L, lianas; V, vines; A, annual and biennial herbs; P: perennial herbs.

were mostly affected by patch-scale factors. However these factors affecting nature- and urban-dependent birds were not similar at the same scale. Habitat quality was the major factor influencing UD birds, and CONTAG also mainly affected their richness. The UD abundance was higher in the tea plantations at the south, and was negatively affected by slope, patch density, and maximum patch index. ND diversity was mainly influenced by CONTAG. The species composition of plants in the tea plantations also had a positive effect on richness and abundance, whereas the patch density, maximum patch index, habitat quality, and percentage of tea plantations in the buffer zone had a negative effect. However, slope did not affect the bird diversity in these tea plantations, which was beyond our expectations.

## DISCUSSION

### Tea plantations provide a new habitat option for birds in the artificial landscape

Our results show that both ND and UD birds were present in the tea plantations, indicating that tea plantations can provide a habitat for different birds. The proportion of ND birds

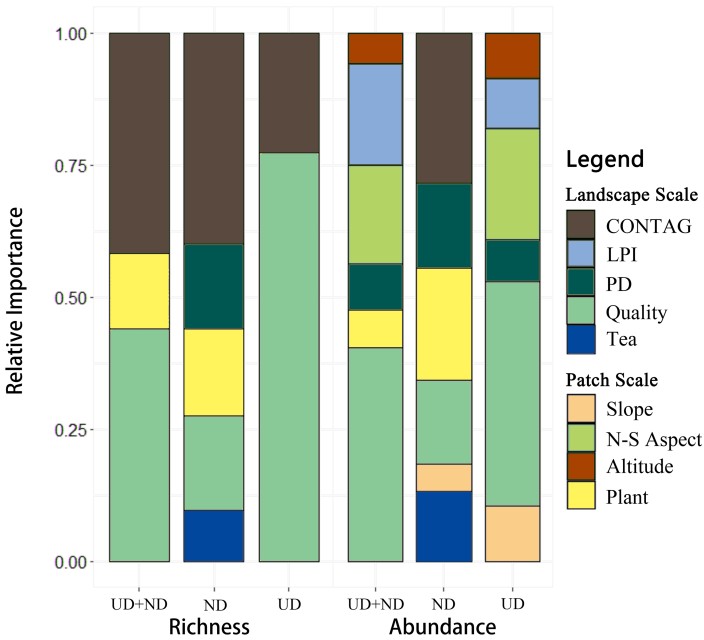

**Figure 5 Effects of explanatory variables on the recorded bird guilds.** ND, nature-dependent birds; UD, urban-dependent birds; Altitude, average altitude; CONTAG, degree of agglomeration or extension of patch types in the buffer zone; LPI, area share of the largest patches in the buffer zone; N-S Aspect, cosine function of slope orientation; PD, number of patches per unit area in the buffer zone; Plant, number of species in the plot; Quality, weighted average of the areas of different types of land around the tea plantation patches in the buffer zone; Slope, average slope; Tea, proportion of tea plantation patches in total patch area in the buffer zone.

in the tea plantation was higher than that of UD birds, presumably because tea plantations retain more natural properties than the surrounding artificial land (*Liu, Fan & Zhou, 2018*). Most tea plantations transformed from forests have reserved higher natural characteristics and native plants than other converted land types. For example, tea plantation shows the highest value for both soil C and N stocks compared with the original mixed-dipterocarp rain forest and Kekilla fernlands, and Caribbean pine plantations (*Sohng, Singhakumara & Ashton, 2017*). Therefore, tea plantations can provide a refuge space for ND birds displaced by forest destruction and urban expansion, playing the role of corridors connecting the isolated ecotone zones, and a temporary and safe place for birds. Meanwhile, tea plantations were attractive to UD birds and could be considered an extra resource or refuge to compensate for the consequence of human activities in urban areas. This supports the finding of previous studies that open green spaces in cities had a positive effect on birds (*Yang et al., 2015*; *Lee et al., 2019*). In addition to the attractiveness of lawns to tourists and urban exploiters (*Paker et al., 2014*), and as the food source of farmlands with highly disruptive (*Lee, Chen & Zou, 2022*), tea plantations play a greater role in bird conservation, because of their close-to-natural characteristic and lower human disturbances (*Halfwerk & Oers, 2020*). For example, compared to urban recreational functions, cannot support long-term visitors. Although some management parks and other open entertainment

**Table 2 Best fitted GLM models used to explain the relationship between bird diversity and environmental and landscape variables.** P-value was adjusted by Bonferroni correction.

| Bird guild | Dependent variable | | Forest plot | Effect | SE | Z-value | P-value | Df |
|---|---|---|---|---|---|---|---|---|
| | | | | | | | | |
| | | | **Independent variable** | | | | | |
| Completion | Richness ($R^2$=0.453) | Plant | | 0.120 | 0.083 | 1.446 | 0.445 | 1 |
| | | CONTAG | | -0.302 | 0.105 | -2.868 | 0.012 | 1 |
| | | Quality | | -0.288 | 0.106 | -2.721 | 0.020 | 1 |
| | Abundance ($R^2$=0.319) | Altitude | | 0.158 | 0.075 | 2.091 | 0.219 | 1 |
| | | Plant | | 0.106 | 0.070 | 1.511 | 0.784 | 1 |
| | | PD | | -0.241 | 0.115 | -2.102 | 0.213 | 1 |
| | | LPI | | -0.340 | 0.115 | -2.952 | 0.019 | 1 |
| | | Quality | | -0.391 | 0.095 | -4.126 | <0.001 | 1 |
| | | N-S Aspect | | -0.210 | 0.069 | -3.022 | 0.015 | 1 |
| Nature-Dependent | Richness ($R^2$=0.166) | Plant | | 0.261 | 0.131 | 1.987 | 0.235 | 1 |
| | | PD | | -0.485 | 0.188 | -2.579 | 0.050 | 1 |
| | | CONTAG | | -0.611 | 0.194 | -3.144 | 0.008 | 1 |
| | | Quality | | -0.595 | 0.230 | -2.585 | 0.049 | 1 |
| | | Tea | | -0.330 | 0.182 | -1.813 | 0.349 | 1 |
| | Abundance ($R^2$=0.515) | Plant | | 0.301 | 0.115 | 2.617 | 0.053 | 1 |
| | | PD | | -0.552 | 0.171 | -3.231 | 0.007 | 1 |
| | | CONTAG | | -0.704 | 0.187 | -3.767 | 0.001 | 1 |
| | | Quality | | -0.778 | 0.42 | -3.223 | 0.008 | 1 |
| | | Tea | | -0.474 | 0.172 | -2.750 | 0.036 | 1 |
| | | Slope | | 0.224 | 0.142 | 1.587 | 0.688 | 1 |
| Urban-Dependent | Richness ($R^2$=0.429) | CONTAG | | -0.392 | 0.158 | -2.079 | 0.075 | 1 |
| | | Quality | | -0.483 | 0.161 | -2.996 | 0.005 | 1 |
| | Abundance ($R^2$=0.402) | Altitude | | 0.312 | 0.098 | 3.193 | 0.008 | 1 |
| | | PD | | -0.211 | 0.141 | -1.493 | 0.813 | 1 |
| | | LPI | | -0.316 | 0.142 | -2.220 | 0.159 | 1 |
| | | Quality | | -0.608 | 0.126 | -4.822 | <0.001 | 1 |
| | | Slope | | -0.152 | 0.105 | -1.451 | 0.880 | 1 |
| | | N-S Aspect | | -0.308 | 0.086 | -3.562 | 0.002 | 1 |

spaces, tea plantations, characterized by fewer practices and harvest were carried out on tea plantations, they only occurred for a limited number of days (mostly one month in Spring) and with low frequency. In addition, most tea plantations are located close to built-up areas in East and South China because of the rapid urbanization in these regions. Tea plantations around severely disturbed built-up areas can be considered an appropriate compensation for the lack of green space, which is necessary for the cluster, breeding, and other behaviors of UD birds (*Šálek, Riegert & Grill, 2015*). Thus, tea plantation is relatively weak in terms of human interference and can, therefore, be regarded as a transitional zone between natural forests and urban areas, which support the diversity of ND and UD birds.

### Species composition of vegetation affects the habitat selection of bird guilds

Analysis of bird-plant association showed that both nature- and urban-dependent birds were strongly correlated with trees and shrubs regardless of the presense of tea. This indicates that birds prefer tea plantations with richer vertical structures and more stable composition of vegetation, that the findings on the positive effect of abundant shade trees and shrubs on bird diversity in tea plantations in Sri Lanka (*Hanle, Singhakumara &*

*Ashton, 2021*). Perennial herbs are more attractive to ND birds than annual and biennial herbs because they provide greater shelter with higher height and leaf density (*Lask et al., 2020*), increasing their sense of security. Moreover, the vegetation composed of perennial plants is more natural and stable and has denser growth than annual plants, providing birds with sustainable food and shelter sources (*Raman et al., 2021*). Compared with UD birds, ND birds are less adaptable to anthropogenic disturbances. Appropriate retention and planting of native plants in tea plantations may further restore forest habitats before degradation and offer the ND birds a familiar habitat for survival in the forest. UD birds preferred the tea plantations with more annual and biennial herbs, and their community composition was associated with major types of plants. This suggests that the distribution of UD birds was more general for the habitat characteristics, which was due to their greater habitat adaption, disturbance tolerance, and various feeding patterns. Overall, the appeal of tea plantations as bird habitats is influenced by the plant compostion, and ND birds preferred the tea plantation with more woody plants and long lifespan herbs, while the habitat preference of UD birds was more generalized.

## Habitat loss and fragmentation caused birds inhabiting tea plantations

The results of GLM show the landscape attributes, as landscape agglomeration and habitat quality were the dominant factors affecting bird richness, which meant the habitat selection of birds actually depended on the landscape pattern surrounding the target tea plantation. Birds choose the tea plantation as their habitat because of the habitat loss and fragmentation of the surrounding landscape. The drivers of habitat loss and fragmentation in Anji are shifting from initial agricultural expansion to current urbanization. Many native birds are disappearing in survival habitat fragments because of the area and isolation effects of habitat fragmentation, which eventually lead to the homogenization of biomes and their functions (*Arasa-Gisbert, Arroyo-Rodríguez & Andresen, 2021*). Tea plantations surrounded by intensive human activities could be treated as relatively intact agroforestry complex ecosystems that preserve some characteristics of the original habitat and provide some birds with the necessary resources to survive in constrained environments (*Exantus, Beaune & Cézilly, 2021*). Tea plantations are an appropriate habitat or stepping stone for birds in an artificial landscape. Additionally, the susceptibility of the population size and community composition of birds in tea plantations to environmental conditions varies across spatial scales. Species richness was mainly influenced by landscape-scale metrics, while abundance was influenced by the joint effects of patch-scale factors. Besides vegetation composition, the north-south aspect of the tea plantations was an important metric for bird abundance. The aspect could affect the microclimate and moisture content of plants (*Wei, He & Li, 2021*), which probably affects the quality of food and habitat for birds (*Perez-Ordonez et al., 2022*). Thus, the landscape configuration around tea plantations was the main factor driving bird preference for tea plantations, while the habitat quality of the tea plantations was the direct factor determining whether the physiological behavior of birds could be successfully maintained.

## Sustainable development of tea plantations

Tea plantations are relatively complex agroforestry ecosystems and have higher biodiversity value than other artificial open spaces. To fully utilize the ecological potential of tea plantations and realize their economic value, it is important to consider tea plantations as a transitional state between natural and artificial ecosystems, support sustainable urbanization, and provide suitable habitats for species under high stress in cities. The idea of sustainability entails integrating the biophysical circumstances of tea plantations into planning and designing management processes and considering the community's human resources. To maintain sustainability in a given area, the proposed site should be built in a way that supports the balance between human needs and the natural environment (*Ahern, Cilliers & Niemelä, 2014*)). Studies have shown that proper retention and planting of native shade trees in tea gardens could enhance the vertical structure of plant communities, effectively regulate the internal microclimate, and substantially support the restoration of biodiversity on tea plantations (*Tscharntke et al., 2011*). By utilizing a complex forest-tea structure or rehabilitating a portion of the surrounding forests, the output of tea plantations could increase with higher bird diversity (*Torralba et al., 2016*). Additionally, minimizing the use of pesticides on tea plantations improved plantation resilience and stability while optimizing the retention of natural elements (*Wang et al., 2022*). Therefore, landscape planning for tea plantations should be based on the concept of maximizing their ecological potential and minimizing negative disturbances in protected areas. The design of vegetation should consider the topography and microclimate of the planning site as well as the surrounding landscape to construct a food-rich and safe habitat for birds in the cities. Additionally the planning and design of tea plantations should support the conservation of the land's natural features, increase the resistance and stability of plantations, and prevent vegetation and soil degradation, to increase the habitat amount for both nature- and urban-dependent birds, which also could improve tea production and quality.

## CONCLUSION

Our findings highlight the role of tea plantations in maintaining bird diversity, which benefits both nature- and urban-dependent birds. We found that the landscape-scale factors surrounding the tea plantations mainly affected bird richness due to habitat selection. Landscape agglomeration and habitat quality were the dominant landscape-scale metrics. The patch-scale factors of tea plantations, especially the vegetation structure, mainly affected bird abundance. Nature-dependent birds preferred to inhabit tea plantations with perennial herbs, whereas urban-dependent birds were attracted to general distributed plants, as annual herbs. Based on our findings, we recommend that the planning and design of tea plantations should consider the inner characteristics of the tea plantation as well as the surrounding landscape. To guarantee the effectiveness of tea plantations for bird conservation, we suggest that native trees should be retained properly in tea agroforestry to increase the food and refuge supply. In addition, environmental-friendly management practices in the tea plantation, instead of the pesticides and other chemicals, could promote the development of ecological tea plantations and urban sustainability.

### Funding

This work was funded by the International Collaborative Project of National Key R & D Plan: 2018YFE0112800, the National Natural Science Foundation of China (32171570), Fundamental Research Funds of Zhejiang Sci-Tech University (2021Q036), and First Class of Disciplines-B of Zhejiang Province (Civil Engineering). The funders had no role in study design, data collection and analysis, decision to publish, or preparation of the manuscript.

### Grant Disclosures

The following grant information was disclosed by the authors:
International Collaborative Project of National Key R & D Plan: 2018YFE0112800.
National Natural Science Foundation of China: 32171570.
Fundamental Research Funds of Zhejiang Sci-Tech University: 2021Q036.
First Class of Disciplines-B of Zhejiang Province (Civil Engineering).

### Competing Interests

The authors declare there are no competing interests.

### Author Contributions

- Jueying Wu conceived and designed the experiments, performed the experiments, analyzed the data, prepared figures and/or tables, authored or reviewed drafts of the article, and approved the final draft.
- Jinli Hu performed the experiments, analyzed the data, prepared figures and/or tables, and approved the final draft.
- Xinyu Zhao performed the experiments, analyzed the data, prepared figures and/or tables, and approved the final draft.
- Yangyang Sun performed the experiments, prepared figures and/or tables, and approved the final draft.
- Guang Hu conceived and designed the experiments, performed the experiments, analyzed the data, authored or reviewed drafts of the article, and approved the final draft.

### Data Availability

   The raw measurements are available in the Supplemental Files.

### Supplemental Information

Supplemental information for this article can be found online at http://dx.doi.org/10.7717/peerj.14801#supplemental-information.

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
