# Peer review of "Role of tea plantations in the maintenance of bird diversity in Anji County, China"

_PeerJ, doi:10.7717/peerj.14801_

## Round 0.1 · original submission · Major Revisions

Thank you very much for your fine contribution to the journal. Please work on comments and resubmit it afterward. Please take help from some English-speaking colleagues as the paper has language mistakes. Secondly, please try to add the recent literature on the topics as the length of the paper is short and use a uniform
reference style as per journal requirements.

·

Basic reporting

Refer to my comments

Experimental design

Survey is conducted nicely. Needed to write the models in the methods and material section. For further, see my comments in attached file

Validity of the findings

Results are derived nicely however not elaborated in write way. Needed to focus more on Table 2 in both methodology and result and discussion section.

Additional comments

The paper in the present form is not suitable for publication. The authors should review the following
items:


Abstract: Method and methodology (survey in your case) needed to add in the abstract. Keywords needed to add. It will increase the accessibility of the published paper.
Your Abstract needs more specific. I suggest that you remove the “mammals” and others as you are not doing research on this (line-19). Be specific to birds.
Introduction:
Line 37: try to use US dollar or write exchange rate USD to Yuan so that an international reader can better understand the monetary value of it.
There is need to add the relationship between tree plantation and bird diversity which is rather scarce in the introduction.
Methods:
Accept my appreciation for writing/conducting good survey. However, empirical method you used in result and discussion needed to add in this section e.g. Poisson-linked generalized linear model and others if any.
Results: Results are presented poorly. Needed more detail. Try to elaborate each variable and justify from past literature. You need to elaborate Table 2 in more detail. This is econometric model which neither discuss in methodology nor discuss in Results and Discussion.
Discussion: Rather to review past studies better to justify your results from this discussion. Readers do not need to read the reviews of past studies. Rather, they expect how you justify your result from past studies. This will further strengthened or reject the existed theory or theories. In nutshell, your objectives are partially met.
Conclusion: Missing. Write the effective conclusion.

Dr Ghulam Mustafa (Assistant Professor)

Reviewer 2 ·

Basic reporting

- Line 115: the hyphen in the last word was unnecessary.
- Line 241: “buildup” should be replaced with “built-up”.
- Some spaces between two words were missing, such as “tourban-dependent” in line 236 and “isnecessary” in line 249.
- There are too many errors in the reference format, please correct. For example, line 337-338: JOURNAL OF THE ROYAL STATISTICAL SOCIETY; line 343: ((2)); line 346 :31(September); line 361:(Camellia sinensis) plantation; line 372: tea (Camellia sinensis) plantations; line 374-375: punctuation; line 399: punctuation; line 401: AVIAN CONSERVATION AND ECOLOGY; line 421-423; line 448; line 450; line 453 and so on.

Experimental design

- What was the data used to calculated the landscape metrics? DEM data was only referred to the topography, not involved the land use information. More detail about the land use data should be provided, including the source and resolution.
- More information about the bird classification was necessary. How did you estimate which bird species preferred the urban or natural habitat?
- I found the author used different words, as “composition”, ”group”, “guild”, which were confused. I suggested to use “guild”.

Validity of the findings

- It is not clear what the different performance of bird diversity between the tea plantation and other alternative land use type. Some comparison with urban green spaces or cropland would help to understand the contribution of the tea plantation.
- Figure 1 Where did the map come from? The National Geographic Information Resources Catalog service system is available on the official website of the Ministry of Natural Resources, PRC. Please download the map from the official website and record the number.
- Figure 4 Why are 0, 0.3, 0.1 and so on taken as the cut-off point? Please state your reasons.
- The figures in your paper are a bit blurry. Please consider replacing them with high-solution ones.Such as Figure 5 is not clear enough.

Reviewer 3 ·

Basic reporting

The language of the manuscript is not up to mark and needs some editing before going further review. The references’ formatting is not uniform, and it should be as per journal’s guidelines.

Experimental design

The experimental design is well. However, authors should be more transparent in presenting data and methods.

Validity of the findings

The findings are well-described. However, presentation of the results may be improved. Moreover, some diagnostic should be performed for validity of the results. Using only Z-stat is not very reasonable choice.

Additional comments

The literature survey is missing, which could help in tracing literature gap and contribution of the study. The direct suggestions out of results are missing. The future direction of the study would be helpful to readers. The conclusions of the study are not well-traced and need to be presented to summarized the major findings of the study.

---

## Round 0.2 · Minor Revisions

Please add the statistical instruments of VIF and ANOVA and re-submit the paper.

·

Basic reporting

Paper is improved now

Experimental design

experiment id improved but still statistical instrument of VIF and ANOVA needed to add. I do not why authors did not include? Is there any rebuttal?

Validity of the findings

findings are improved now

Additional comments

no further comments

Reviewer 3 ·

Basic reporting

All comments has been taken into consideration. The paper is accepted for publication.

Thank you.

Experimental design

.

Validity of the findings

.

Additional comments

.

---

## Round 0.3 · accepted · Accept

All reviewers' comments have duly been incorporated and the paper is accepted for publication.